# Optimizing Sample Size for Population Genomic Study in a Global Invasive Lady Beetle, *Harmonia Axyridis*

**DOI:** 10.3390/insects11050290

**Published:** 2020-05-09

**Authors:** Hongran Li, Wanmei Qu, John J. Obrycki, Ling Meng, Xuguo Zhou, Dong Chu, Baoping Li

**Affiliations:** 1College of Plant Protection, Nanjing Agricultural University, Nanjing 210095, China; Hongran.Li@uky.edu (H.L.); ml@njau.edu.cn (L.M.); 2Key Lab of Integrated Crop Pest Management of Shandong Province, College of Plant Health and Medicine, Qingdao Agricultural University, Qingdao 266109, China; register1_kq@163.com (W.Q.); chinachudong@qau.edu.cn (D.C.); 3Department of Entomology, University of Kentucky, Lexington, KY 40546, USA; john.obrycki@uky.edu (J.J.O.); xuguozhou@uky.edu (X.Z.)

**Keywords:** sample size, *Harmonia axyridis*, 2b-RAD, empirical investigation, population genomics, genetic diversity

## Abstract

Finding optimal sample sizes is critical for the accurate estimation of genetic diversity of large invasive populations. Based on previous studies, we hypothesized that a minimal sample size of 3–8 individuals is sufficient to dissect the population architecture of the harlequin lady beetle, *Harmonia axyridis*, a biological control agent and an invasive alien species. Here, equipped with a type IIB endonuclease restriction site-associated (2b-RAD) DNA sequencing approach, we identified 13,766 and 13,929 single nucleotide polymorphisms (SNPs), respectively, among native and invasive *H. axyridis* populations. With this information we simulated populations using a randomly selected 3000 SNPs and a subset of individuals. From this simulation we finally determined that six individuals is the minimum sample size required for the accurate estimation of intra- and inter-population genetic diversity within and across *H. axyridis* populations. Our findings provide an empirical advantage for population genomic studies of *H. axyridis* in particular and suggest useful tactics for similar studies on multicellular organisms in general.

## 1. Introduction

Using optimal sample sizes to accurately estimate genetic diversity of large natural populations is an imperative issue in the analysis of evolutionary processes [1]. Because larger samples per population than needed for accurate estimation, results in extra expense and wastes much time to analysis [2], while limited sample sizes will lead to significant errors in estimating the genetic diversity of species [3,4,5]. Still, most genetic studies of wild species sampled so many individuals per population, but using a small number of genetic markers, for example, microsatellites [5,6,7,8]. In practice, the number of microsatellite markers is often limited owing to the cost and time, and thus lowers the power in addressing phylogeographic questions [5,9,10]. Using microsatellites markers to analyze population genetics of invasive species could be a concern, since a recent study showed that low genetic diversity or genuine multimodality was observed [11].

Given these disadvantages of microsatellites marker in genetic analysis, a growing number of researches have recommended to use the genomic data over microsatellites for genetic study [12,13,14]. For example, a recent empirical study showed that genome-wide techniques can acquire large numbers of single nucleotide polymorphisms (SNPs), to obtain a finer population structure and stronger patterns of isolation-by-distance than microsatellites do with a smaller sample size [15]. It is gratifying that recent developments in restriction site-associated DNA sequencing (RAD-seq) techniques, could provide massive sequence data for efficient identification of SNPs at an unprecedented level [16,17,18]. Nevertheless, although the costs associated with next-generation sequencing have reduced substantially, RAD-seq remains a relatively costly approach due to the uncertainty of the number of individuals needed to be sampled in a given study. Therefore, a trade-off between the sample size and the number of molecular markers must be considered during the experimental design.

To our knowledge, the utilization of these new techniques has effectively addressed the establishment of an ideal sampling scheme for sample size determination [19,20,21,22]. For example, an empirical study via double digest RAD-seq technique showed that the genetic diversity and *Fst* in a plant species can be accurately estimated from six to eight individual plants [1]. A simulation analysis using a large number of SNPs has shown that sample size can be reduced to four to six individuals when estimating the genetic differentiation (*Fst*) [2]. Using a type IIB endonucleases RAD-seq method, Qu et al. [23] have provided the optimization of sampling schemes for an invasive whitefly species, *Bemisia tabaci* Gennadius, which showed that a sample size greater than four individuals has little impact on estimates of genetic diversity. However, the sampling space is limited, which might increase the overlaps between iterations [1].

The harlequin lady beetle, *Harmonia axyridis* (Pallas) (Coleoptera: Coccinellidae), has emerged as an alien invasive species in many North America and European countries [24,25,26,27,28]. Inferences about introduction routes of this invasive species are needed to understand the fundamental eco-evolutionary aspects of colonization success or failure [29] and for preventing future invasions [30]. Using microsatellite loci data, the approximate Bayesian computation (ABC) methods with more traditional statistical approaches have been reconstructed the introduced routes of *H. axyridis* worldwide, as two bridgehead invasive populations were involved in North America and then served as the source populations for at least six independent introductions into other continents [31]. However, there is no research to characterize the population genomics of *H. axyridis* worldwide based on as many as SNP markers, which may provide novel insights in genetic structure and introduced routes. Indeed, the prerequisite is to determine effective sample size from individual populations to accurately estimate population genetic parameters. Following the results from the existing empirical studies in other organisms, we hypothesized that a minimal sample size of 3–8 individuals is sufficient to estimate genetic diversity within and across native and invasive *H. axyridis* populations, although other factors can also contribute to population genetic inferences [1,23].

To test this hypothesis, we conducted the experiments as follows: (1) surveyed SNPs among *H. axyridis* populations from both native and invasive ranges using type IIB endonuclease RAD sequencing (2b-RAD) method, (2) constructed simulated populations using a random subset of individuals, and finally (3) determined the minimal number of individuals to accurately estimate the several intra- and inter-population genetic diversity parameters, including number of effect alleles (*Ae*), observed heterozygosity (Ho), unbiased expected heterozygosity (*uHe*), and pairwise genetic differentiation (*Fst*) in *H. axyridis*; the ad hoc statistic ΔK was supplemented to determine the optimal sample sizes for intra- and inter-population genetic diversity running the STRUCTURE software [23,32].

## 2. Materials and Methods

### 2.1. Harmonia Axyridis Collection and DNA Extraction

A total of 20 *H. axyridis* females per population were collected from China (LNSY), its native range (41.83°N, 123.57°E) and Poland (PLKK), its invasive range (50.06°N, 19.93°W) for subsequent sequencing analysis. The specimens were preserved in 95% ethanol and stored at −80 °C until DNA extraction. Total DNA was extracted using a TIAMamp Micro DNA Kit (Tiangen Biotech (Beijing) Co., Ltd., Beijing, China) following the manufacture’s protocols. Extracted DNA was dissolved in DNAase-free water and DNA concentration was determined using NanoDrop 2000 spectrophotometer (Thermo Fisher Scientific, Waltham, MA, USA). DNA integrity was assessed using 1.0% agarose gel electrophoresis.

### 2.2. Library Preparation and SNP Identification

Briefly, 2b-RAD libraries were constructed for each individual *H. axyridis* following Wang et al. [33]. Subsequently, genomic DNA was digested by 1 U BsaXI (New England Biolabs, cat. no. R0609), and short adapter sequences were ligated to the ends of the fragments. The ligation products were amplified in 50 µL PCRs; only fragments starting with a sequence that can be hybridized by the selective sequence of the primer was efficiently amplified. PCR products were purified using a MinElute PCR Purification Kit and pooled for sequencing using the Illumina PE sequencing platform.

Raw sequence data were filtered as follows: (1) the reads with linker sequences were removed to obtain clean reads; and (2) reads with low-quality positions (>15% of nucleotide positions with a Phred score < 30) were deleted. In N bases greater than 8% and without restriction recognition sites, the filtered high-quality sequences were referred to as enzyme reads. Then mapping the enzyme Reads to the *H. axyridis* reference genome (http://bipaa.genouest.org/sp/harmonia_axyridis) was done using SOAP program (the parameter was set to: -r0-M4-v2) and the same reads were clustered into Unique Tags. Finally, the SNP-calling was performed using Maximum likelihood (ML) method [34]. To ensure the accuracy of SNP genotyping, the following filtering procedures were performed: (1) SNPs with a minor allele frequency (MAF) < 0.01 were deleted; (2) tags with more than 2 SNPs were deleted; (3) SNPs at each locus with 1 or 4 bases were deleted; and (4) SNPs that could be genotyped in more than 80% of the individuals were retained.

### 2.3. Population Genetic Analyses

The allelic data generated by 2b-RAD sequencing were used to build a neighbor-joining (NJ) phylogenetic tree between LNSY (China) and PLKK (Poland) populations. Phylogenetic reconstruction was carried out in treebest v.1.9.2, with 1000 bootstrap replicates [35]. ADMIXTURE (v.1.3.0) was carried out using the entire dataset to estimate the genetic ancestry of each sample [36]. This tool is based on a maximum likelihood approach, which provides an estimation of the number of genetic clusters and the proportion of derived alleles in one sample from each of the K populations. The program was run 10 times, varying the values of K from 2 to 10. A cross-validation test was performed to determine the optimal value of K. The results from 10 replicates of the selected K values were summarized into a single result and were then aligned and analyzed using pong, a network-graphical approach for analyzing and visualizing membership in latent clusters. The raw data have been deposited in the Sequence Read Archive (SRA) database under an accession number of SRP227109.

### 2.4. Construction of Simulated Populations

Before we proceeded to the optimal sample size, we carried out a power analysis to determine the minimum number of resampling replicates to ensure accurate estimation of genetic parameters following Qu et al. (2019) [23] with minor modifications. Specifically, GenALEx 6.5 was used to remove SNPs deviating from Hardy–Weinberg equilibrium for each population [37]. Then, SNPs that potentially under balancing and divergent selection were removed using the software BAYESCAN v. 2.1 with 20 pilot runs of 10,000 iterations, a burn-in of 50,000 iterations and a final run of 100,000 iterations [1]. To reduce the false positives, prior odds of the neutral model were set to 100 (i.e., the neutral model is 100 times more likely than the model with selection). After then, a total of 3000 SNPs (k) were randomly selected using data tools in Excel for each population. Considering that too limited sampling space would increase overlaps between iterations, we enlarged sampling space in the data set (*n* = 20) than that in previous study (*n* = 10) [23]. In detail, we constructed simulated data sets consisting of different numbers of resampling replicates (x = 10, 20, 30, 40, 50, 60, 70, 80, 90, 100), each represented by the different sample sizes (number of individuals per population, *n* = 2, 4, 6, 8, 10, 15).

### 2.5. Optimizing Sample Size

GenALEx 6.5 was used to measure the genetic parameters, including the number of effective alleles (*Ae*), observed heterozygosity (Ho), and unbiased expected heterozygosity (*uHe*), for each replicate at each sample size (i.e., for each simulated population) [37]. To estimate the degree of genetic differentiation, i.e., pairwise genetic differentiation (*Fst*) among populations, a slightly different subsampling strategy, was used to resample the 3000 SNPs shared by LNSY and PLKK populations rather than LNSY or PLKK individually. In addition, we used an ad hoc statistic, ΔK, based on the rate of change in the log probability of data between successive K values, to evaluate the optimal number of replicates and sample sizes for population genomics analyses [32]. ΔK shows a clear peak at the optimal number of replicates and sample sizes.

GraphPad Prism 7.00 was used to measure the influence of sample sizes and replicates on intra- and inter-population genetic diversity parameters.

## 3. Results

### 3.1. Population Characterization and SNP Identification

Sequencing of the 2b-RAD libraries resulted in about 344.31 million raw reads from the 40 individuals. On average, 8.61 million reads with restriction site per individual were retained. After quality filtering, the percentage of high quality of reads was above 80% (305.57 million enzyme reads were generated) of the total reads in the libraries of the 40 individuals. Of the retained sequences, a total of 104.56 million (34.26%) enzyme reads aligned to the *H. axyridis* genome survey sequences (https://bipaa.genouest.org/sp/harmonia_axyridis/) (Appendix A). Of these, 1.67 million (57.92%) loci with minimum 3X and maximum 500X coverage were retained for SNP discovery (Appendix A). After removing SNPs that significantly deviated from HWE (818 for population LNSY and 655 for population PLKK), we identified 13,766 and 13,929 polymorphic SNPs for further analysis (Appendix A). We did not detect any loci that were under selection for either population of *H. axyridis*, with the false discovery rate (FDR) set to 0.05. As such, no other loci were removed from subsequent analyses (Appendix A). The number of effective alleles (*Ae*) in the *H. axyridis* populations was 1.123 ± 0.001 SE (LNSY) and 1.124 ± 0.001 SE (PLKK). The expected heterozygosity (He) in the LNSY and PLKK populations was 0.087 ± 0.001SE and 0.087 ± 0.001SE. The observed heterozygosity (Ho) in the LNSY and PLKK populations were 0.076 ± 0.001SE and 0.079 ± 0.001SE, respectively. The *Fst* calculated from all detected SNPs distances between LNSY and PLKK populations was 0.036, indicating that no substantial genetic differentiation exists between the two populations.

### 3.2. Population Genetic Structure

The resultant neighbor-joining (NJ) tree of these ladybug populations featured two main clusters, namely LNSY and PLKK (Figure 1). The analysis of population structure estimated the K value was two, indicating that the uppermost hierarchical level detected by STRUCTURE was two distinct genetic clusters (Figure 2).

### 3.3. Depicting Sample Sizes for Intrapopulation Genetic Diversity

We assessed the impact of increasing sample sizes for intrapopulation genetic diversity estimates by resampling 2, 4, 6, 8, 10, and 15 individuals from empirical data sets obtained for the two *H. axyridis* populations. At first, accurate estimates of population genetic parameters in two populations were acquired in our simulations with only x = 30 resampling replicates (Figure 3 and Figure 4). In detail, when we fixed the number of individuals (*n*) to three and the number of SNPs (k) to 3000, there was no statistical difference for the mean values of *Ae*, Ho and *uHe* even when the number of replicates was set to x = 30 [LNSY: *Ae* = 1.1047, 95%CI (1.1015, 1.1079); Ho = 0.0767, 95% CI (0.0738, 0.0796); *uHe* = 0.0802, 95% CI (0.0775, 0.0829). PLKK: *Ae* = 1.1112, 95%CI (1.1091, 1.1133); Ho = 0.0803, 95% CI (0.0786, 0.0820); *uHe* = 0.0845, 95% CI (0.0832, 0.0858)] or x = 100 [LNSY: *Ae* = 1.1006, 95%CI (1.0970, 1.1042); Ho = 0.0731, 95% CI (0.0703, 0.0759); *uHe* = 0.0771, 95% CI (0.0745, 0.0797). PLKK: *Ae* = 1.1121, 95%CI (1.1089, 1.1153); Ho = 0.0803, 95% CI (0.0781, 0.0825); *uHe* = 0.0848, 95% CI (0.0825, 0.0871)]. At the same time, the ΔK line chart showed a peak at x = 30 (Appendix A).

Our simulations allowed us to determine the minimum sample size of *H. axyridis* required to ensure that the sample precisely reflects the genetic diversity of the empirical data sets. In the LNSY population, increasing sample sizes above four (*n* ≥ 4) individuals appears to have little impact on the mean *Ae*, Ho and *uHe* when 3000 SNPs were selected for *n* = 4 [*Ae* = 1.1071, 95%CI (1.1056 1.1085); Ho = 0.0764, 95% CI (0.0750, 0.0777); *uHe* = 0.0805, 95% CI (0.0793, 0.0817)] and for *n* = 15 [*Ae* = 1.1097, 95%CI (1.1093, 1.1102); Ho = 0.0758, 95% CI (0.0754, 0.0762); *uHe* = 0.0807, 95% CI (0.0804, 0.0811)] (Figure 5). Also, at *n* = 4, the ΔK line chart showed a clear peak (Figure 5). For the PLKK population, sample sizes above six individuals appear to have little impact on the mean Ho, when 3000 SNPs were considered. The mean values of Ho for *n* = 6 was 0.0795 [95% CI (0.0788, 0.0802)] and for *n* = 15 was 0.0795 [95% CI (0.0792, 0.0799)]. Simultaneously, the ΔK line chart showed a clear peak at *n* = 6 (Figure 6). For *Ae* and *uHe* parameters, a small sample size (*n* = 4) with 3000 SNPs was sufficient to recover the genetic diversity found in PLKK populations [for *n* = 4: *Ae* = 1.1149, 95% CI (1.1139, 1.1159); *uHe* = 0.0848, 95% CI (0.0842, 0.0855); for *n* = 15, *Ae* = 1.1185, 95% CI (1.1182, 1.1189) and Ho = 0.0849, 95% CI (0.0847, 0.0852)] (Appendix A). The ΔK line chart shows a clear peak at *n* = 4, separately (Figure 6).

### 3.4. Determination of the Sample Sizes for Interpopulation Genetic Diversity

To estimate the degree of population genetic differentiation, our results showed that compared to x = 100, there was no statistical difference for the mean values of *Fst* when we set the number of replicates to x = 60; the number of individuals (*n*) were fixed to three and the number of SNPs were fixed to 3000. For instance, the mean values of *Fst* for x = 60 were 0.0398 [95% CI (0.0265, 0.0531)] and for x = 100 were 0.0347 [95% CI (0.0286, 0.0408)] (Appendix A). At the same time, the ΔK line chart showed a peak at x = 60 (Figure 7). Furthermore, increasing sample size above six (*n* ≥ 6) did not decrease the sample size needed to recover the genetic differentiation of individual populations of *H. axyridis* based on 60 replicates (Figure 8). The mean values of *Fst* for *n* = 6 were 0.0370 [95% CI (0.0345, 0.0396)] and for *n* = 15 were 0.0394 [95% CI (0.0388, 0.0400)]. Similarly, the ΔK line chart showed a peak at *n* = 6 (Appendix A).

## 4. Discussion

In this study, we conducted a rigorous empirical determination of the optimal sample size in an invasive ladybeetle, *H. axyridis*, which confirmed our hypothesis that a minimal sample size of 3–8 individuals is sufficient to estimate genetic diversity within and across native and invasive populations. The next step in our studies will be to sample global populations of *H. axyridis* to investigate the genetic diversity. To our knowledge, only one published study has examined a reliable number of individuals to address invasive population genomics using SNP markers [23], while several studies have investigated the impact of sample size based on microsatellite markers [5,8,38,39,40,41].

Sample size is an important study design factor that influences population genomic studies [1,15,23,42,43]. There are generally two types of sampling that occur in invasive genomic studies. First, there is process variance, due to variations in genetic metrics caused by the number of individuals introduced, the diversity and differentiation of the source population(s), multiple introductions, genetic drift, and natural dispersal [44,45,46]. Second, there is sampling variance, caused by variation in allele frequencies when a subset of individuals (the sample) is drawn from the population [1,23,44].

Our results indicate that, in general, even small sample sizes are likely to be sufficient, which is similar to previous studies [1,15,23,42,43]. Specifically, we found that four to six individuals were enough to calculate within-population genetic diversity estimates using RADseq that provides a large number of SNPs. These results are inconsistent to previous study of another invasive whitefly *B. tabaci*, in which 3–4 individuals were required to recover within-population genetic diversity parameters (Table 1). The exact reason for this phenomenon is unclear here, but may be associated with the invasive process variance of two invasive species (e.g., bottleneck effects and founding effects occurred in different periods for these species, causing different extent of gene flow, etc.). Another interpretation may be our enlarged sampling space in the data set (*n* = 20) than that in previous study (*n* = 10) [23], which could deliberately decrease overlaps between iterations. Actual and recommended sample sizes for evaluating a population have varied widely. For example, previous studies have used a larger data set (*n* = 25–30 or 30 individuals) from each population to identify the optimal sample size [43,47], and 35 individuals were evaluated in previous study [1]. In our study, we removed SNP markers showing deviation from Hardy–Weinberg equilibrium and those under selection for each population of *H. axyridis* to estimate genetic parameters [1], which should not be overlooked to explain the different results with Qu et al. [23].

Additionally, our results showed that sample sizes of six individuals per population, provide accurate estimates of *Fst*, when a large number of polymorphic SNPs are employed. This minimum sample size in *H. axyridis* is larger than two individuals for an Amazonian plant species, *Amphirrhox longifolia* (Violaceae) and three to four individuals for *B. tabcai* based on empirical analysis [1,23] (Table 1). However, in a simulation study, the population sample size reported can be as small as *n* = 4–6 to measure *Fst* metrics when using a large number of SNPs (>1000) [2]. It is worth noting that 3000 SNPs as described in Qu et al. 2019 [23] is sufficient for reliable estimation of genetic diversity parameters, which is greater than the number used in previous studies to estimate *Fst* (>1000 or ≥1500 SNPs) [1,2]. However, a different study used 23,057 SNPs to determine optimal sample size for the Galapagos tortoise [42], while another empirical simulation study employed approximately 14,000 SNPs [43]. Thus, we confirmed that this resampling method is effective and robust, but it may be necessary to assess the appropriate sample size for each invasive species prior to the characterization of their invasion genetics. The main reason is these estimates can be affected by many factors in evolutionary process including the bottleneck effect, founder effect, and bridgehead effect [46].

In present study, we selected native (Asian) and invasive (European) populations of *H. axyridis* to conduct a sample size optimization analysis. The SNP data support the presence of two highly divergent lineages of *H. axyridis* populations. Lineage 1 is distributed in China and Lineage 2 generally in Poland, which suggests a long-standing barrier to gene flow between these geographic regions (Figure 1 and Figure 2). Our results revealed that similar sample size could accurately estimate the genetic metrics of the two populations, which indicate that the optimal sample size of *H. axyridis* is not dramatically affected by the invasion process.

## 5. Conclusions

Our results showed that a sample size greater than six individuals (*n* ≥ 6) has little impact on the estimates of genetic diversity within *H. axyridis* populations. Accurate estimates of *Fst* can also be easily obtained at a small simple size (six individuals). The findings demonstrated that SNP markers can accurately estimate the genetic diversity of *H. axyridis* populations, even when small numbers of individuals are sampled, which provides a starting point for future genome-wide population studies.

## Figures and Tables

**Figure 1 insects-11-00290-f001:**
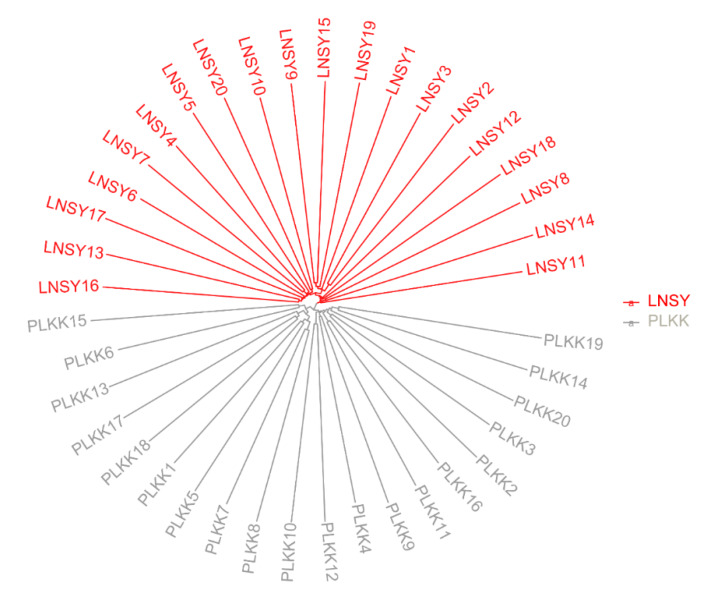
Neighbor-joining phylogram of the LNSY and PLKK populations of *H. axyridis*. The different colors represent the individuals from different populations.

**Figure 2 insects-11-00290-f002:**
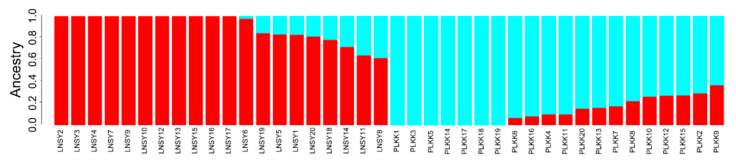
Admixture for LNSY and PLKK populations of *H. axyridis* (K = 2). Each bar represents an individual from each of the collection locations (X axis). Individual admixture coefficients are represented in each column (Y axis).

**Figure 3 insects-11-00290-f003:**
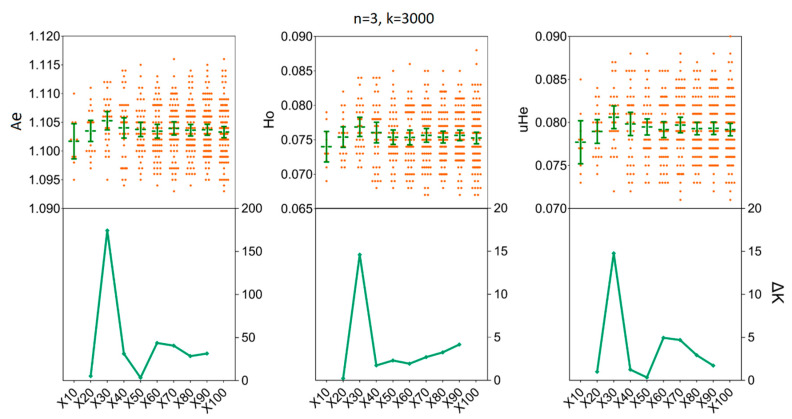
The minimum number of resampling replicates (x) required to estimate the genetic diversity in LNSY population. Bars represent sample means and 95% confidence intervals of the means. The ΔK (*Y*-axis) shows a clear peak at the optimal replicates (x). *Ae*, number of effective alleles; Ho, observed heterozygosity; *uHe*, unbiased expected heterozygosity.

**Figure 4 insects-11-00290-f004:**
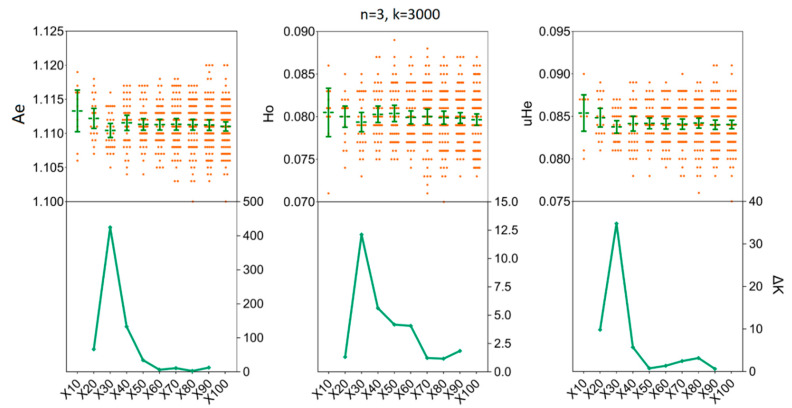
The minimum number of resampling replicates (x) required to estimate the genetic diversity in PLKK population. Bars represent sample means and 95% confidence intervals of the means. The ΔK (*Y*-axis) shows a clear peak at the optimal replicates (x). *Ae*, number of effective alleles; Ho, observed heterozygosity; *uHe*, unbiased expected heterozygosity.

**Figure 5 insects-11-00290-f005:**
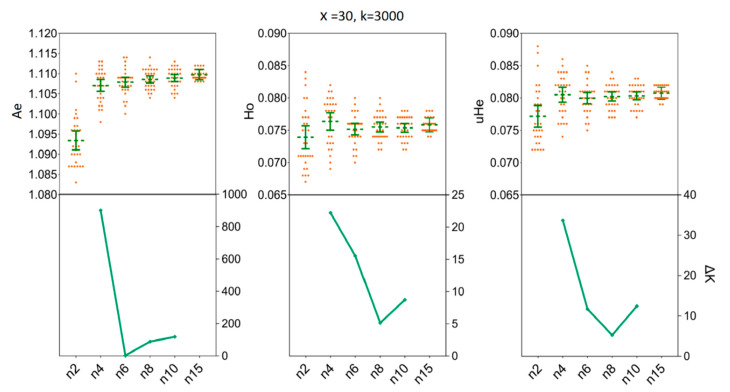
The minimum sample size (*n*) required to estimate the genetic diversity in LNSY population. Bars represent sample means and 95% confidence intervals of the means. The ΔK (*Y*-axis) show a clear peak at the minimum sample sizes (*n*). *Ae*, number of effective alleles; Ho, observed heterozygosity; *uHe*, unbiased expected heterozygosity.

**Figure 6 insects-11-00290-f006:**
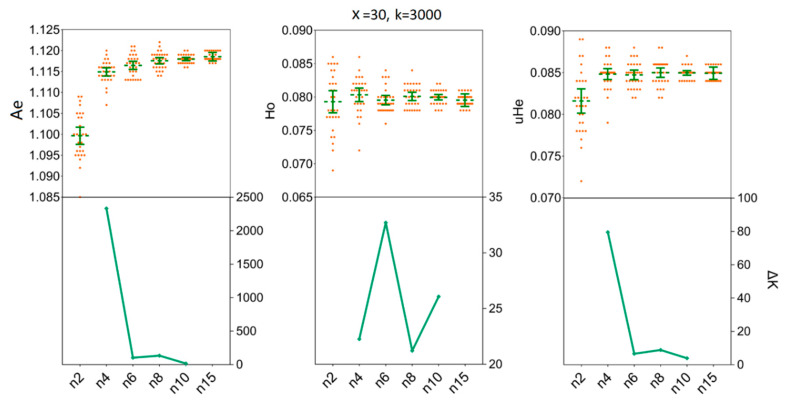
The minimum sample size (*n*) required to estimate the genetic diversity in PLKK population. Bars represent sample means and 95% confidence intervals of the means. The ΔK (*Y*-axis) shows a clear peak at the minimum sample sizes (*n*). *Ae*, number of effective alleles; *Ho*, observed heterozygosity; *uHe*, unbiased expected heterozygosity.

**Figure 7 insects-11-00290-f007:**
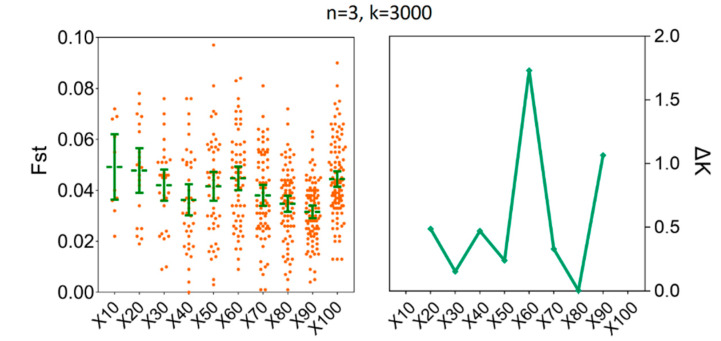
The minimum number of resampling replicates (x) required to estimate *Fst* between LNSY and PLKK populations. Bars represent sample means and 95% confidence intervals of the means. The ΔK (*Y*-axis) shows a clear peak at the optimal replicates (x). *Fst*, pairwise genetic differentiation.

**Figure 8 insects-11-00290-f008:**
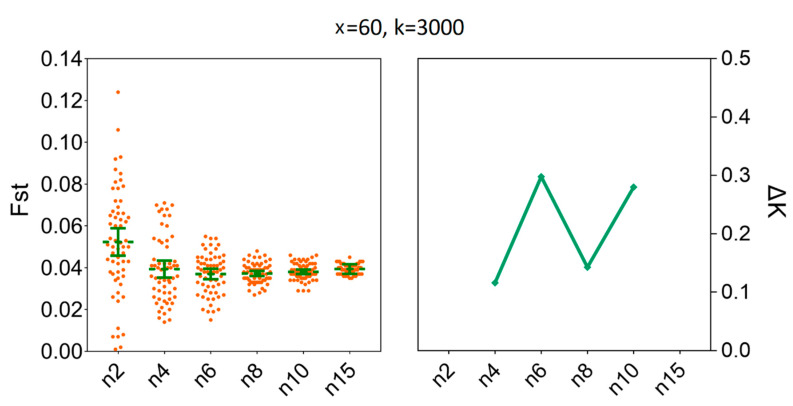
The minimum sample size (*n*) required to estimate *Fst* between LNSY and PLKK populations. Bars represent sample means and 95% confidence intervals of the means. The ΔK (*Y*-axis) shows a clear peak at the minimum sample sizes (*n*). *Fst*, pairwise genetic differentiation.

**Table 1 insects-11-00290-t001:** The optimal sample sizes required for different genetic parameters.

Species Analyzed	Genetic Diversity			Genetic Differentiation	References
	Ae	Ho	uHe	(Fst)	
*Bemisia tabaci* MED Q1 clade	3	3–4	3	4	Qu et al. 2019
*Bemisia tabaci* MED Q2 clade	3–4	4	3–4	3	Qu et al. 2019
*Amphirrhox longifolia*	2	2	6–8	2	Nazareno et al. 2017
*Harmonia axyridis*	4	4–6	4	6	The present study

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
