# Peer review of "Optimizing Sample Size for Population Genomic Study in a Global Invasive Lady Beetle, *Harmonia Axyridis"

_insects, 2020, doi:10.3390/insects11050290_

Round 1
Reviewer 1 Report
Dear Authors:
Several corrections to the text are necessary.
Line 116: "used to build"
Line 174: lady beetle
Line 281: Bemisia tabaci and give authority name
Line 297: Give authority name for species when first mentioned in the text. Please correct spelling; B. tabaci and add italics.
Line 300: Que et al. 2019
Line 312: "suggests"
Line 313: "accurately"
Line 322, 323: Abbreviate scientific name - H. axyridis
Line 448: Add italics to scientific name
Figure Legends (all figures): Please abbreviate scientific name - H. axyridis.
Author Response
Dear Editor and reviewer,
I am pleased to receive such a prompt response from the reviewers on our manuscript. I, after discussions when necessary with other authors, have made a revision to it, with a special reference to reviewers’ comments. All changes are explained below in response to the comments point by point (in red color). In addition, I have made some changes (with the traces left for your reference) to the Abstract to straighten it up.
Thanks for your help in making our manuscript improved.
With best wishes,
Baoping
Baoping LI,
at Nanjing Agricultural University, Nanjing.

Reviewer 2 Report
First of all, congratulations on this good job. The authors show how modern molecular biological methods and only a few individuals from an insect population can be used for complete genomical characterization of the population. Although the authors have already shown this in a previous work on another insect species, the Asian ladybird species examined in the current study is particularly interesting because it is an invasive species worldwide and little is known about its genetic diversity. A surprising result is that the two populations from China and Poland examined here differ only slightly genetically.
The experiments were carried out carefully and the results are clearly described.
I only have a few linguistic corrections:
- line 51: a trade-off was firstly faced...
- line 54: scheme, not schemes
- line 92: total DNA was extracted...
Author Response

(The authors gave the same response as above.)
